# Using Augmented Reality to Enhance Students' Representational Fluency: The Case of Linear Functions †

**Shuhui Li** [1] , **Yihua Shen** [2], **Xinyue Jiao** [3] **and Su Cai** [3,4,*]

1   School of Mathematical Sciences, East China Normal University, Shanghai 200241, China; shli@math.ecnu.edu.cn
2   Mathematics, Science, and Technology Department, Teachers College, Columbia University, New York, NY 10027, USA; ys2866@tc.columbia.edu
3   VR/AR+ Education Lab, School of Educational Technology, Faculty of Education, Beijing Normal University, Beijing 100875, China; 202021010198@mail.bnu.edu.cn
4   Beijing Advanced Innovation Center for Future Education, Beijing Normal University, Beijing 100875, China
*   Correspondence: caisu@bnu.edu.cn
†   This study belongs to a series of studies on using Augmented Reality to enhance students' learning of liner functions. We collected the students' self-efficacy in mathematics in pre-tests and post-questionnaires, which were excluded in this paper due to different research focus.

**Abstract:** Using multiple representations is advocated and emphasized in mathematics and science education. However, many students have difficulty connecting multiple representations of linear functions. Augmented Reality (AR) may affect these teaching and learning difficulties by offering dynamically linked representations. Inspired by this, our study aims to develop, implement, and evaluate an AR-based multi-representational learning environment (MRLE) with three representations of linear functions. The data were collected from 82 seventh graders from two high-performing classes in an urban area in China, through a pre-test, a post-questionnaire, and follow-up interviews. The results reveal that students were satisfied with the AR-based MRLE, which assisted in enhancing their understanding of the real-life, symbolic, and graphical representations and connections among them. Regarding students' interactions with multiple representations, apparent differences in learning sequences and preferences existed among students in terms of their representational learning profile. In sum, learning in the AR-based MRLE is a complex interaction process between the mathematics content, forms of representations, digital features, and students' representational learning profile.

**Keywords:** augmented reality; representational fluency; linear function; multiple representations; STEM

**MSC:** 97U10

## 1. Introduction

The existing literature and many reforms in mathematics education emphasize the role of representations in mathematics teaching and learning [1–3]. *Representational fluency*, the ability to work within and translate among multiple representations, is vital to developing students' conceptual understanding and problem-solving skills [4–6]. However, prior research revealed that many K-12 students exhibited representational disfluency. For example, one of the biggest stumbling blocks for algebra students was translating between various representations of functions [7]. New technology, which offers linked representations, may provide solutions to those teaching and learning difficulties [8].

Augmented Reality (AR), a burgeoning technology that overlays virtual objects into the real world, brings significant changes to educational settings and has become an essential focus of research partly due to the accessible and affordable hardware [9]. Previous studies have shown that AR has demonstrated its strength in sustaining students' spatial

thinking and conceptual understanding of abstract mathematics concepts [10]. Additionally, using AR, multiple linked representations can be presented simultaneously, convenient for graphing and calculating. These findings open up the possibility of using AR as an innovative tool for empowering students' representational fluency.

Therefore, this study aims to develop an AR-based multi-representational learning environment, examine how students learn different representations of linear functions (a traditional core topic in secondary school mathematics) in such an environment, and produce insights into viable approaches to enhance students' representational fluency.

*1.1. Multiple Representations in Linear Functions*

Representations are defined differently in different domains (epistemology, educational psychology, etc.). In mathematics education, researchers differentiated between internal and external representations and assumed that connections between internal unobservable mental representations are influenced and stimulated by building connections between corresponding external representations [11]. Thus, in this study, we focus on external representations, which include spoken language, symbols, pictures or graphs, concrete or computer-based manipulative models, etc. [5,12]. These representations are important in their own right, and, more importantly, flexible connections among them play a critical role in learning mathematics, especially in helping students build a deep conceptual understanding and develop flexibility in problem solving [11–15]. However, abundant studies showed that secondary or even college students often had difficulty making connections among representations, especially among tabular, graphical, verbal, and symbolic representations of linear functions [16–20], which may be related to the restriction of representations used in teaching or the curriculum [17,21].

A much-debated question is: "Are two representations better than one?" Research to date has not yet determined whether Multi-Representational Learning Environments (MRLE) result in superior mathematics learning. Researchers argue that multiple representations have additional learning benefits since different representations can complete each other, and one representation may constrain the interpretation of the other [22]. On the other hand, the split-attention and redundancy effects [23] indicated that multiple representations might contain redundant information and increase cognitive loads on a student's cognitive system [24], as more information must be processed simultaneously and stored in working memory. It may lead to students' failure in performing the task when the sum of cognitive loads exceeds the working memory limits [25].

Additionally, empirical studies on the learning effect of MRLE usually indicated inconclusive results. For example, Kolloffel et al. compared the effect of using Diagram (D), Arithmetic (A), Text (T), T + A, and D + A and showed that T + A was the most beneficial for learning combinatorics problems [26]. While in Rolfes et al.'s study [27], Table + Graph leads to advantages in learning qualitative functional thinking, but Graph leads to higher gains in quantitative functional thinking. They indicated that the learning effect is a complex interaction between learning content and representations.

Function, ubiquitous in our everyday life, is considered one of the most critical and fundamental contents for mathematics education, in which multiple linked representations are widely used [8,16]. Moschkovich et al. identified five common representations of a function: equation, table, graph, verbal description, or real-world situation [28]. When students use various representations of functions, they are almost forced to look at functions in three main aspects: *mapping, covariation*, and *function as object*, which is the most common approach for describing functional thinking [29]. The first aspect *mapping* means that for every element x of the domain, there is exactly one element y of the range; the second aspect *covariation* focuses on the variation of the independent variable and the resulting covariation of the dependent variable; the third aspect *function as object* takes the whole function into account [30].

Researchers have examined how the aforementioned three aspects occur in different representations of functions. A great deal of previous relevant research examined only one

or two aspects, emphasizing the covariational aspect [19,27], and few studies covered all three aspects in various representations simultaneously. Moreover, relevant research usually focused on the graph, table, verbal description, and algebraic expression of functions and connections among them [6,15,21,28], while real-world situations received less attention, even though the use of real-life situations is expected to stimulate students' engagement in the process of learning.

The Emergent Modelling design [31] and the literature on realistic mathematics education [32] stressed the importance of starting with contextual problems to develop situation-specific reasoning and tentative representations. In addition, many algebra students failed in translating from a word problem or its verbal description to an algebraic expression or graph [7]. Therefore, there is a need for more research integrating real-world situations to examine the potentially positive role of real-life problems in representational learning and offer students engaging and meaningful learning opportunities.

Drawing on the above literature review, we established the conceptual framework for this study (see Figure 1), focusing on connections among real-life (RL), symbolic (S), and graphical (G) representations of linear functions in three aspects: *mapping*, *covariation*, and *function as object*, for the following reasons. First, abundant published studies [16–20] described students' learning difficulty in connecting G<->S of linear functions. Meanwhile, G and S are the most prevalent and prototypical representations when teaching functions [33]. Second, based on the Integrated Model of Text and Picture Comprehension (ITPC) [34], the information processing of descriptive (such as symbolic) and depictive (such as graphical) representations occur via different channels. Data on how students learn with representations from different channels would be helpful for the MRLE design. Third, considering the importance of linear functions in secondary mathematics, rich real-life applications of linear functions in STEM subjects [35,36], and related learning difficulties [7], this study attended to the linear function and its real-life situations.

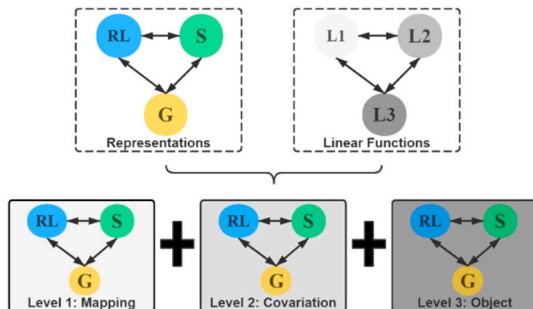

**Figure 1.** Conceptual framework of representational fluency in linear functions.

Moreover, individual characteristic is a factor influencing a student's representational fluency [37], for example, a student's prior knowledge (including misconceptions) about representations [16,38], domain-specific knowledge [39], representational preference, and affective factors [22,40]. In addition, researchers discovered that students still had difficulty translating between representations though they had mastered each representation individually [41]. There exists a call for more empirical investigations on how different students deal with each representation in MRLE [27]. Up to now, far less attention has been given to how students with varied individual characteristics interact with different representations in MRLE. It may produce valuable insights to design effective MRLE and thus help students with various needs build their representational fluency.

In sum, previous studies have outlined considerable difficulties students encounter in developing representational fluency, and the learning effect of MRLE is influenced by several factors, such as content, representation types, and students' characteristics. In the meantime, new technology, which makes dynamically linked multiple representations available in MRLE [8], offers new possibilities to conquer students' learning difficulties.

### *1.2. Using Augmented Reality to Enhance Students' Mathematics Learning*

In recent years, an increasing number of educators and researchers have advocated using dynamic software to support representation and function learning. For example, Ocal [42] and Zulnaidi et al. [43] demonstrated that GeoGebra could facilitate students' conceptual and procedural understanding of function-related topics, such as limit and derivative, and promote interactive learning. Moreover, researchers reported that Geometer's Sketchpad could help students visualize and understand the graphs of different functions and improve their attitude toward learning [44,45].

All these dynamic software packages possess a common beneficial feature, i.e., when learners operate on one representation, they will see the corresponding effects on all other representations, namely *dynamic linking* or *automatic translation* [46]. Scaife and Rogers [47] proposed that such a feature can reduce the cognitive load placed on students and enable them to concentrate on the study of the relation between representations. However, previous studies often focused on the symbolic, graphical, and tabular representations of a function, leaving verbal and real-world situated representations unexplored. An underlying reason could be the inability to provide technical support for other forms of representations, for example, a real-world environment.

Augmented Reality (AR), a new technology that displays computer-generated virtual information next to the physical objects that reside in a real-world environment, could be a promising solution to the current problem. Researchers have already shown that AR provides a unique and meaningful platform for teaching and learning in the STEM fields [48]. Ahmad and Junaini [49] conducted a systematic review of 19 journal articles between 2015–2019 concerning studying mathematics in an AR-based environment and generalized three main themes: implementation, development, and effectiveness of AR. In addition, they reported three major benefits of AR: (1) increasing students' self-confidence and mathematical understanding, (2) enhancing students' visualization skills, and (3) promoting interactive learning.

It is noticeable that the research topics and findings pertaining to AR were similar to those concerning dynamic software, probably because they all share dynamic and automatic characteristics. Researchers illustrated that AR could reduce students' efforts in tedious tasks, such as drawing functions manually and performing algebraic calculations, allowing them to concentrate on more critical assignments, such as exploring and analyzing [50]. Moreover, what differentiates AR from other dynamic software is that it can present virtual information in close spatial proximity to real objects, thus expanding the research domain regarding representations. Additionally, Altmeyer et al. [51] pointed out that this closeness between virtual information and actual objects satisfies Mayer's twelve design principles for multimedia learning [52], which are valuable for reducing learners' extraneous cognitive load, fostering generative processing, and supporting conceptual knowledge acquisition. Overall, AR not only exhibits the strength of dynamic software but also provides new research directions, such as integrating real-life situations. This study presumed that an AR-based MRLE could potentially enhance students' understanding of linear functions' real-life, symbolic, and graphical representations.

Moreover, we reviewed the existing literature regarding the use of AR to support representation and function learning and discovered that the application of AR in algebra is relatively scarce compared to that in geometry. Existing studies have focused on interactive learning [53,54], spatial visualization skills [50,55], and the affective domain [56,57]. Consequently, this study seeks to offer new insights into how AR can be utilized to facilitate the teaching and learning of three representations of linear functions.

Last but not least, Cai et al. [58] noted that empirical studies on using AR in mathematics classrooms often focused on students' learning gains and motivations, but only a few have addressed the difference between students with different personal features. Consequently, they conducted a study to evaluate how an AR app influenced students with different levels of self-efficacy. Another research study conducted by Chen [59] revealed that learners with high anxiety exhibited higher confidence and satisfaction than those with

lower anxiety in an AR-based mathematics course. We reviewed more relevant articles, for example [60], and discovered that these studies often categorized students based on non-mathematical characteristics. However, few have targeted students' specific mathematics knowledge, such as their representational knowledge, which may produce insights into the future design of MRLE for personalized learning of mathematics.

### 1.3. Research Questions

The purpose of this study is to develop, implement, and evaluate an AR-based MRLE, which provides rich learning opportunities for students to work with dynamically linked multiple representations of real-life motion problems. To be specific, the AR games focus on three representations, real-life(RL), symbolic (S), and graphical (G), targeting six types of connections: RL->S, RL->G, S->RL, S->G, G->RL, and G->S of linear functions in three aspects (*mapping*, *covariation*, and *function as object*). Our overarching research question is: What contributions could AR bring to students' representational fluency in linear functions? More specifically, this study is guided by the following research questions:

1. How do middle school students perceive the role of an AR-based multi-representational learning environment in their learning of linear functions?
2. How do middle school students interact with representations in an AR-based multi-representational learning environment? Do differences exist between students with varied representational learning profiles?

By addressing these questions, we hope that this study could provide research evidence to shed light on issues regarding students' learning processes in MRLE and advance our understanding of the role of AR in enhancing students' representational fluency.

## 2. Materials and Methods

### 2.1. Targeted Mathematics Topics

In this study, we focused on the application of linear functions in the uniform linear motion problem, a key topic in secondary school physics and mathematics [61]. Based on the conceptual framework, this study targeted concepts in three aspects of functional thinking: *mapping*, *covariation*, and *function as object* [30] and their corresponding real-life, symbolic, and graphical representations (see Table 1). Students were expected to connect three representations of each concept listed below.

**Table 1.** Mathematics concepts and representations tackled in this study.

| Aspect | Concept | Real-Life (RL) | Symbolic (S) | Graphical (G) [1] |
|---|---|---|---|---|
| Mapping | A pair of values satisfying a linear function | Departure<br>Destination | $(t_0, S_0)$<br>$(t_1, S_1)$ | Initial point<br>End point |
| Covariation | Rate of change | Speed: run (fast/slow) or stop | $v$: positive (big/small) or 0 | Slope: Steepness (steep/gradual) or horizontal |
| Function as Object | Linear function | One player travelling on one track with a constant speed or taking a break | $S = v\,(t - t_0) + S_0$ | Line |
| | Constant function [2] | One player taking a break | $S = S_1$ | Horizontal line |
| | Piecewise linear function | One player travelling on one track with breaks et al. | $S = \begin{cases} vt\ (t_0 \leq t \leq t_1) \\ S_1\,(t_1 \leq t \leq t_2) \end{cases}$ | More than one line |

[1] Due to a typesetting issue, graphical representations are shown in Table 2. [2] A special case of linear functions.

*2.2. AR App and the Intervention Design*

In line with the above mathematics concepts and representations, the AR app for Android tablets was developed using Unity 3D and Vuforia, which consisted of two games: (1) Game 1: Let's go hiking and (2) Game 2: AR$^+$ Beijing travel plan. The app interface includes three dynamically linked representations: a 3D animation of the real-life motion problem, the corresponding graphs, and algebraic expressions.

Table 2 lists the motion problem context, stages of play, app interface, and underlying mathematics in detail. In Game 1, a student first chooses one vehicle from four vehicle cards to complete his/her trip from home A to mountain B, then chooses the time planned for hiking, and finally reaches the top of mountain B. In Game 2, a student can choose one vehicle for his/her trip from place A to B, playtime in B, another vehicle for his/her trip from place B to C, playtime in C, a new vehicle for his/her trip from place C to D, and playtime in D, and finally finish his/her own Beijing travel plan.

To maximize students' learning effectively, we consulted the 12 principles to structure multimedia material listed in the Cognitive Theory of Multimedia Learning when designing the app [52]. To reduce extraneous load, we included only graphics and text that support learning goals and simple visuals (Coherence principle), used fireworks as signals to draw attention to important information (Signaling principle), placed text close to the graphics it refers to (Spatial Contiguity principle), and presented corresponding animations, symbols, and graphs simultaneously (Temporal Contiguity principle). To manage intrinsic load, we adopted the Segmenting principle, which Game 1 is the simplified version of Game 2, aiming to provide students with enough scaffolding support to be engaged in the game and make the mathematics easier. A review of key concepts and representations in linear functions, as well as a teachers' demonstration of how to play the game, were also offered to students before they play (Pre-training principle). In addition, we used conversational language in the app (Personalization principle).

The intervention module consists of two sessions. The first session contains a 10 min review of concepts and representations in linear functions, a 5 min teacher demonstration of Game 1, a 20 min students' self-exploration of Game 1, a 5 min classroom discussion, and a 5 min summary. The second session has an 8 min teacher–students co-play time of Game 2, a 22 min students' self-exploration of Game 2, a 10 min classroom discussion, and a 5 min summary. Students are encouraged to observe and compare three dynamically linked representations while playing games. In addition, corresponding worksheets for each session are provided to students to ensure necessary support when exploring mathematics in the AR-based MRLE.

**Table 2.** Interfaces of the AR-based MRLE.

| AR Game | Motion Problem | Stage | AR App Interface (Sample) | Underlying Mathematics |
|---|---|---|---|---|
| Game 1: Let's go hiking | [8:00 am at home A], you plan to go to mountain B for hiking. G Map app recommends 4 vehicles to get there, namely: bicycle, bus, subway, and taxi. Now, you can choose your vehicle card and begin your trip. | Stage 1: Choose one vehicle. |  | |
| | | Stage 2: Choose the time planned for hiking. |  |  |
| | | End: Reached the top of mountain B. |  | $S = vt \quad (0 \le t \le t_1)$ $S = S_1 \quad (t_1 < t \le t_2)$ |

**Table 2.** *Cont.*

| AR Game | Motion Problem | Stage | AR App Interface (Sample) | Underlying Mathematics |
|---|---|---|---|---|
| Game 2: AR$^+$ Beijing travel plan | The 24th Winter Olympic Games are scheduled to open in Beijing on 4 February 2022. Our class plans to travel to Beijing together. Our leading teacher will reserve a hotel near Tsinghua University (A), then travel to the Bird's Nest (B), then to the Temple of Heaven (C), and finally to the CCTV Headquarters (D). Please choose suitable vehicles and complete your own Beijing travel plan. | Stage 1: Choose one vehicle (A to B). When you arrived at B, choose the playtime in B.<br><br>Stage 2: Choose one vehicle (B to C). When you arrived at C, choose the playtime in C.<br><br>. . .<br><br>End: Finish the Beijing travel plan. |   . . .   | <br>$S = v_1 t$  $(0 \le t \le t_1)$<br>$S = S_1$  $(t_1 < t \le t_2)$<br>$S = S_1 + v_2(t - t_2)$  $(t_2 < t \le t_3)$<br>$S = S_2$  $(t_3 < t \le t_4)$<br>$S = S_2 + v_3(t - t_4)$  $(t_4 < t \le t_5)$<br>$S = S_3$  $(t_5 < t \le t_6)$ |

*2.3. Data Collection Instrument*

The study used a pre-test to evaluate students' prior representational knowledge of linear functions and a post-questionnaire to see students' perceptions of how AR plays a role in their learning. In addition, a follow-up structured interview was conducted to grasp deeper information about students' learning process.

To assess students' prior representational knowledge, we designed the pre-test consisting of three overarching real-life motion problems. The first problem is presented in text, including 3 items on RL->S and 3 items on RL->G. The second problem is presented mainly in algebraic expressions, having 3 items on S->RL and 3 items on S->G. The third problem is shown mainly in a graph, with 3 items on G->RL and 3 items on G->S. 3 items for each connection targets *mapping*, *covariation*, and *function as object* aspects. A sample item (targeting S->G, aspect: *function as object*) is shown in Figure 2.

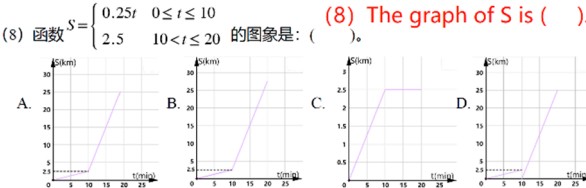

**Figure 2.** One sample question from the pre-test.

To see how students perceive the role of AR in their learning of mathematics, we designed the post-questionnaire with 15 five-point Likert scale items (1 for strongly disagree and 5 for strongly agree), focusing on three dimensions: AR user satisfaction (4 items), AR utility (9 items), and students' learning preference (2 items). Aligned with the questionnaire items, the interview protocol (see Appendix A) was designed to triangulate the questionnaire data and gather more in-depth information about how students learn in an AR-based MRLE.

Two rounds of pilot tests were conducted, one round with two students and the other round with six students (from the same school but not in the sampled classroom). Based on the feedback, we went through multiple rounds of revisions and finally produced the final version. Two experienced mathematics teachers (including the mathematics teacher of the sample classes) and a Ph.D. in mathematics education were invited to review and adjust the written test, which guaranteed the validity of the assessment.

The reliability of the designed tests is established by the internal consistency reliability coefficient [62], which is considered by researchers as one of the most appropriate statistical methods to assess reliability [50,63]. Cronbach's $\alpha$ of the pre-test was acceptable, 0.690, and Cronbach's $\alpha$ of three dimensions of the post-questionnaire (AR user satisfaction, AR utility, and learning preference) were 0.928, 0.951, and 0.768, respectively; all passed 0.7, a general rule of thumb indicating a satisfactory internal consistency [64].

*2.4. Sampling and Data Collection*

The AR-based intervention was conducted in two high-performing classes at a public middle school in an urban city at Shandong, China, in December 2021. We intentionally selected the two high-performing classes as the mathematics tackled in this intervention was to some extent challenging for seventh graders, and students from high-performing classes may utilize the AR potential better than those from ordinary classes.

A total of 87 seventh graders (one class with 43 students, the other with 44 students) participated in this AR-based intervention. Before taking the intervention, the 87 students had just finished their learning of the linear function chapter in the sequence of (1) functions, (2) linear functions, (3) the graph of linear functions, (4) the algebraic expressions of linear functions, and (5) applications of linear functions [65]. We believe it is the perfect time for students to have a new multi-representational learning experience since (1) their knowledge about linear functions and the graphical and algebraic expressions seems to be more solid

than in other moments due to the fresh working memory, and (2) sufficient prior knowledge of each representation is one of the keys to effective learning in a multi-representational learning environment [27]. One of the researchers taught the intervention, mainly guiding students to explore the motion problems and pay attention to three simultaneously linked representations in the AR-based MRLE (see Figure 3).

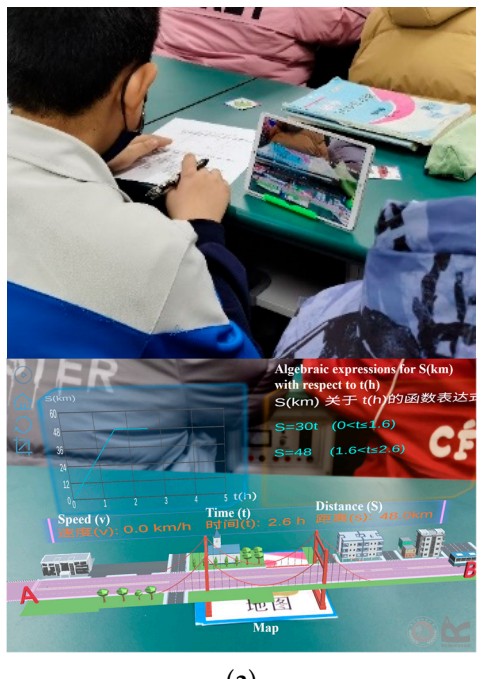
(**a**)

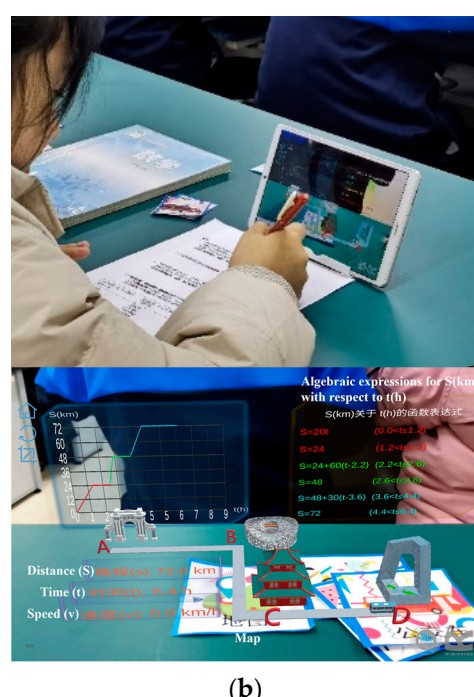
(**b**)

**Figure 3.** Students are playing AR games in the intervention. (**a**) Above: the boy is playing Game 1, below: a screenshot of Game 1; (**b**) Above: the girl is playing Game 2, below: a screenshot of Game 2.

For data collection, all participants had about 30 min to complete the pre-test before the intervention. After the intervention, they finished the post-questionnaire in 10 min. Some students missed one or two tests due to illness or other issues, so we collected back 82 pairs of pre-test and post-questionnaires. The response rate was 94.3%.

Of the 82 participants, 41 were girls, and 41 were boys. Twelve students (ten boys and two girls) were randomly selected from the students who were willing to participate in follow-up interviews: 6 in one-to-one and 6 in paired groups. The average duration was about 9 min. With the students' agreement, all interviews were recorded and transcribed. To keep anonymity, we named 12 interviewees as S1 to S12. For students who were willing to participate in the interview, there were more boys than girls, and thus in the end more boys than girls were randomly selected as interviewees.

### 2.5. Data Coding and Data Analysis

For the pre-test, cluster analysis [66] was used to classify the sample into two groups: high-performing versus low-performing in terms of students' performance in items starting from RL, S, and G, separately. To grasp a more detailed learning profile, we also used cluster analysis to categorize students' performance in six types of connections, C1(RL->S), C2(RL->G), C3(S->RL), C4(S->G), C5(G->RL), and C6(G->S), individually, into high-performing and low-performing based on their score on relevant items. Combining these dimensions, we generated the learning profile depicting students' prior representational knowledge for each student (see Figure 4).

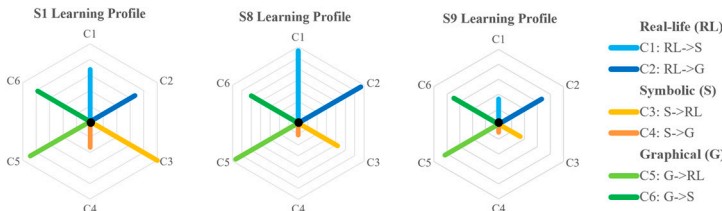

**Figure 4.** Three sample learning profile.

The categorization result is shown in Table 3. A one-sample *t*-test was then conducted to examine students' responses to each dimension for each group and for all students. The results show that each mean value of students' responses to items in one dimension (C1–C6, RL, S, and G) (1) in the high-achieving group was statistically significantly higher than that of the overall group; (2) in the low-achieving group, their responses to the items in one dimension was statistically significantly lower than that of the overall group (see *t*-values in Table 3, *p* < 0.001).

**Table 3.** The clustered students' prior representational performance.

| Group | | C1 | C2 | C3 | C4 | C5 | C6 | RL | S | G |
|---|---|---|---|---|---|---|---|---|---|---|
| High-achieving Group | N | 61 | 67 | 31 | 19 | 60 | 66 | 57 | 22 | 64 |
| | Mean (SD) | 85.24 (16.69) | 72.63 (12.87) | 82.79 (16.23) | 67.25 (14.56) | 85.69 (11.53) | 74.03 (15.82) | 77.38 (11.25) | 73.29 (12.87) | 78.51 (12.71) |
| | *t* | 8.69 *** >overall | 6.12 *** >overall | 12.04 *** >overall | 12.50 *** >overall | 7.98 *** >overall | 5.40 *** >overall | 8.42 *** >overall | 13.37 *** >overall | 6.19 *** >overall |
| Low-achieving Group | N | 21 | 15 | 51 | 63 | 22 | 16 | 25 | 60 | 18 |
| | Mean (SD) | 12.69 (16.58) | 20.00 (17.41) | 26.30 (14.75) | 12.87 (14.69) | 41.41 (16.62) | 20.13 (12.92) | 36.22 (18.01) | 23.10 (12.78) | 33.64 (16.71) |
| | *t* | −14.90 *** <overall | −9.56 *** <overall | −10.33 *** <overall | −6.80 *** <overall | −9.14 *** <overall | −13.41 *** <overall | −7.94 *** <overall | −8.15 *** <overall | −8.88 *** <overall |
| Overall Group (*n* = 82) | Mean | 66.66 | 63.00 | 47.6 | 25.47 | 73.81 | 63.51 | 64.83 | 36.56 | 68.66 |
| | SD | 35.90 | 24.63 | 31.48 | 27.29 | 23.62 | 26.33 | 23.39 | 25.74 | 23.10 |

RL: real-life; S: symbolic; G: graphical; C1: RL->S; C2: RL->G; C3: S->RL; C4: S->G; C5: G->RL; C6: G->S; *t*: one-sample *t* tests; *** *p* < 0.001.

Descriptive statistics, such as means and standard deviations, were used to analyze each Likert scale item in the post-questionnaire. For the interview transcript, as well as the suggestions collected from the open-ended questions in the post-questionnaire, three researchers used Grounded Theory [67] to develop initial codes, discussed inconsistencies until there is an agreement, and finally produced the codebook for answers to each question (see Appendix A). The data were re-examined, and final codes were assigned to each response. To ensure reliability, two researchers coded each transcript according to the codebook independently. The percent agreement between two coders is 99.2%.

## 3. Results

### 3.1. Analysis of the Role of an AR-Based MRLE

#### 3.1.1. The AR User Satisfaction Dimension

Overall, students were highly satisfied with the intervention and the two AR games and would like to have more opportunities to study with AR in the future. The descriptive statistical results are shown in Table 4.

**Table 4.** Descriptive statistical results of students' satisfaction with the AR-based MRLE.

| Dimension | Item Description | Mean | SD | 1 | 2 | 3 | 4 | 5 |
|---|---|---|---|---|---|---|---|---|
| AR user Satisfaction (4 items, *n* = 81) | 1. I am satisfied with the intervention. | 4.73 | 0.548 | 0 | 0 | 4 | 14 | 63 |
| | 2. I am satisfied with the AR Game 1. | 4.72 | 0.597 | 0 | 1 | 3 | 14 | 63 |
| | 3. I am satisfied with the AR Game 2. | 4.73 | 0.548 | 0 | 0 | 4 | 14 | 63 |
| | 4. I want to have more learning opportunities with AR in the future. | 4.70 | 0.660 | 1 | 0 | 3 | 14 | 63 |

According to Table 4, Items 1–4 regarding students' satisfaction obtained mean values of 4.73, 4.72, 4.73, and 4.70 and standard deviations of 0.548, 0.597, 0.548, and 0.660, respectively. Each item's data followed a negatively skewed distribution, indicating more data values are above the mean value. Specifically, 63 (77.8%) students selected "strongly agree", and 14 (17.3%) students chose "agree" for each item, revealing that 95.1% of the students were (highly) satisfied with the intervention.

All students in the interview expressed their satisfaction with the intervention and pointed out that AR games were the most attractive part of the lessons. When asked which AR game was better, three students picked AR Game 1 as they thought the first game was more straightforward, while seven students chose AR Game 2 because it allowed them to compare the representations of different linear functions. S9 said: "The second game contains multiple stops to change the vehicles so that I can compare the representations of different linear functions simultaneously."

Students' responses to the open-ended questions in the post-questionnaire conveyed their suggestions. One third of the students had no suggestions. The remaining two thirds of the students suggested a more updated tablet computer configuration, stable card recognition systems, realistic 3D scene simulation, varied real-life contexts and gameplay, and time for playing and problem solving. The interviews' results also confirmed these recommendations with one additional piece of advice: to optimize the visual effects on the screen, such as the graph color and font size.

3.1.2. The AR Utility Dimension

In general, students were highly satisfied with the utility of AR-based MRLE in deepening their understanding of linear functions within each representation and among them. The descriptive statistical results are shown in Table 5.

**Table 5.** Descriptive statistical results of students' satisfaction with utility of the AR-based MLE.

| Dimension | Item Description | Mean | SD | 1 | 2 | 3 | 4 | 5 |
|---|---|---|---|---|---|---|---|---|
| AR utility in MLE (9 items, *n* = 82) | 5. AR helps me understand real-life motion problems better. | 4.72 | 0.479 | 0 | 0 | 1 | 21 | 60 |
| | 6. AR helps me understand graphs of linear functions better. | 4.79 | 0.437 | 0 | 0 | 1 | 15 | 66 |
| | 7. AR helps me understand algebraic forms of linear functions better. | 4.79 | 0.437 | 0 | 0 | 1 | 15 | 66 |
| | 8. The real-life problem helps me understand its graph better. * | 4.68 | 0.564 | 0 | 0 | 4 | 18 | 60 |
| | 9. The real-life problem helps me understand its algebraic form better. * | 4.67 | 0.546 | 0 | 0 | 3 | 21 | 58 |
| | 10. The graph helps me understand the real-life problem better. * | 4.63 | 0.658 | 0 | 1 | 5 | 17 | 59 |
| | 11. The graph helps me understand the algebraic form better. * | 4.67 | 0.589 | 0 | 0 | 5 | 17 | 60 |
| | 12. The algebraic form helps me understand the real-life problem better. * | 4.61 | 0.643 | 0 | 1 | 4 | 21 | 56 |
| | 13. The algebraic form helps me understand the graph better. * | 4.65 | 0.575 | 0 | 0 | 4 | 21 | 57 |

* In the AR-based multi-representational learning environment.

In Table 5, Items 5–7 received mean values of 4.72, 4.79, and 4.79 and standard deviations of 0.479, 0.437, and 0.437, respectively. The data distribution was negatively skewed for all three items, and 81 (98.8%) students considered AR-based MRLE a valuable setting for comprehending linear functions in real-life, graphical, and symbolic representations separately. During the interviews, nine students selected graphical; eight students picked

symbolic, and only two students chose real-life representations to be the representations that they benefited most from during the AR-based MRLE.

As for Items 8–13, their mean values ranged from 4.61 to 4.68, and standard deviations ranged from 0.546 to 0.658. Similarly, the data distribution was negatively skewed for all six items. The number of students who selected "agree" or "strongly agree" for each item is between 76 (92.7%) and 79 (96.3%), which is slightly lower than those for Items 5–7. Overall, it is safe to conclude that at least 92.7% of the students agreed that the AR-based MRLE could assist them in building connections among three different representations of linear functions.

### 3.2. Students' Interactions with Representations in an AR-Based MRLE
#### 3.2.1. Learning Preference

In brief, students preferred (1) the AR-based to the traditional non-AR and (2) the multi- to the mono-representational learning environment. Table 6 shows the representational learning profile of the twelve interviewees. They were categorized into a high-achieving group (students were in the high-achieving group of at least two dimensions among RL, S, and G) and a low-achieving group (students were only in the high-achieving group of one or no dimensions among RL, S, and G). After the categorization, each group contained six interviewees.

**Table 6.** Students' representational learning profile and their learning sequence.

| Representational Learning Profile | ID | RL | S | G | C1 | C2 | C3 | C4 | C5 | C6 | Learning Sequence |
|---|---|---|---|---|---|---|---|---|---|---|---|
| High-achieving (2H and 3H) | S1 | H | H | H | H | H | H | L | H | H | RL and G->S |
| | S5 | H | H | H | H | H | H | H | H | H | RL->G->S |
| | S6 | H | H | H | H | H | H | H | H | L | RL->G and S |
| | S10 | H | H | H | H | H | H | H | H | H | RL and G and S |
| | S7 | H | L | H | H | H | L | L | H | H | RL->G->S |
| | S8 | H | L | H | H | H | L | L | H | H | RL->G->S |
| Low-achieving (0H and 1H) | S3 | H | L | L | H | H | L | L | L | H | S->G->RL |
| | S4 | H | L | L | H | H | L | L | L | H | G->RL and S |
| | S9 | L | L | H | L | H | L | L | H | H | G->S->RL |
| | S11 | L | L | L | H | L | L | L | H | L | RL and G and S |
| | S2 | L | L | L | L | H | L | L | L | L | G and S->RL |
| | S12 | L | L | L | L | L | L | H | L | H | RL->G->S |

H: high-achieving; L: low-achieving; RL: real-life; S: symbolic; G: graphical; C1: RL->S; C2: RL->G; C3: S->RL; C4: S->G; C5: G->RL; C6: G->S; S2 and S12 prefer mono-representational than multi-representational learning.

Regarding students' learning preference for the AR-based or non-AR learning environment, the item ($n = 79$) acquired a mean value of 4.72 and a standard deviation of 0.505, implying that, on average, students were more inclined to study in an AR-based learning environment. In the interviews, all students preferred the AR-based learning environment to the traditional one. We generalized a list of reasons mentioned by students from each representational learning profile level as follows:

1. For 3H students: AR leads to a more comprehensive and clearer understanding of the concepts and is interesting;
2. For 2H students: AR is interesting;
3. For 1H and 0H students: AR is simple to use, interesting, and helpful in visualizing the concepts.

In sum, we identified three major findings: (1) All levels of students thought AR created a delightful learning environment; (2) The 3H students thought AR enhanced their conceptual understanding of concepts; (3) Low-achieving students (1H and 0H) thought AR helped them to visualize abstract concepts.

In terms of students' learning preference for the multi-representational or mono-representational learning environment, the item ($n = 82$) obtained a mean value of 4.65

and a standard deviation of 0.636, indicating that, on average, students preferred a multi-representational to a mono-representational environment. During the interviews, ten interviewees preferred to study in a multi-representational environment, while two interviewees would rather study in a mono-representational environment. We noticed that these two interviewees exhibited a low-performing representational learning profile in all three dimensions (RL, S, and G), which could probably be the reason behind their choice. In the same way, we summarized students' reasons for their preferences, and the results are shown in Table 7:

**Table 7.** Interview results of students' preference for Multi- or Mono-RLE.

| Dimension | Profile | Mean |
|---|---|---|
| Multi-RLE | 3H | MRLE clarifies the confusion and provides a comprehensive structure. |
| | 2H | MRLE saves problem-solving time and enhances understanding. |
| | 1H | MRLE saves problem-solving time and enhances understanding. |
| | 0H | MRLE saves problem-solving time and provides a comprehensive structure. |
| Mono-RLE | 0H | Mono is easy to understand. |

For the underlying reasons, S1 said: "When we studied representations one by one in the past, I often confused a linear functions' graph with a direct variations' equation. If we could study all representations of linear functions simultaneously, I would not get confused." We simplified this quote into "clarifies the confusion." S8 mentioned: "When we studied representations one by one, it was hard to build connections among them. In this lesson, I built a more comprehensive knowledge structure of linear functions." We shortened this quote to "comprehensive structure." S7 responded: "The understanding of one representation will complement the understanding of another representation, thus, reinforcing the understanding of each other." We abbreviated this quote to "enhances understanding." In summary, students who preferred MRLE praised it for allowing them to make comparisons among different representations and, as a result, deepened their understandings of linear functions.

Although students from all different representational learning profiles expressed their preference for the MRLE, high-achieving students (3H and 2H) and low-achieving students (1H and 0H) exhibited some differences. In the interviews, high-achieving students indicated that they could compare three representations simultaneously, whereas low-achieving students implied that they could only compare at most two representations simultaneously. Last but not least, students who preferred a mono-representational learning environment said they could only work on one representation at a time.

During the interviews, students also specified many advantages and disadvantages of the AR-based MRLE. Based on the grounded theory analysis results, we summarized advantages, such as the AR-based MRLE (1) demonstrated connections between different representations clearly (7 students); (2) promoted a comprehensive structure of multiple representations (4 students); (3) made it easy to calculate/draw and the problem-solving process easier (3 students); and (4) deepened the understanding of functions and function problems in real life (2 students). Regarding disadvantages, seven students mentioned that they felt it hard to concentrate on three representations simultaneously, and two students said they paid too much attention to the real-life animation model.

### 3.2.2. Learning Patterns

Generally speaking, students exhibited eight different learning paths when studying in AR-based MRLE. According to Table 6, the path RL->G->S appears four times; the path RL and G and S simultaneously appears two times, and the rest appears only once. Three interesting patterns can be identified in high-achieving students' learning paths:

1.  All paths follow the sequence: RL->G->S;
2.  All paths start with RL and end with S;

3. Some paths involve processing multiple representations simultaneously.

Regarding the first pattern, students used the words "real" and "realistic" to describe the real-life contexts and "complex" and "hard to imagine" to depict algebraic expressions in the interviews. Therefore, we safely concluded that their learning sequences seem to follow the pattern "Concrete->Semi-concrete->Abstract", which fits the sequence used in the concreteness fading instructional approach [68]. As for the second pattern, RL, S, and G can be further decomposed into C1+C2, C3+C4, and C5+C6, respectively. Looking into students' performances in these subcategories revealed that high-achieving students (3H and 2H) are most proficient in C1 and C2 (RL), then C5 and C6 (G), and finally C3 and C4 (S), which indicates that high-achieving students (3H and 2H) seem to learn from their most proficient ones to less proficient ones. For the third pattern, S1 and S6 were able to process two representations, and S10 could process three representations simultaneously.

On the other hand, low-achieving students' learning paths seem random and disordered. One explanation for this is that low-achieving students have not developed an organized and rigorous learning methodology for themselves. Thus, they need more teachers' guidance to help them progress in an MRLE.

## 4. Discussion

We have outlined three major findings, and each warrants further discussion.

First, it is evident that students who participated in the intervention overall were highly satisfied with the AR-based MRLE, put forward helpful and valuable suggestions for the app upgrade, and looked forward to more AR-based learning experiences. This finding was consistent with much previous research using AR to teach other mathematics topics, for example, geometry [69,70], calculus [54,55], and probability [10]. We can see that AR can be successfully used to teach mathematics topics other than geometry, the current mainstream research domain. Future studies could explore the integration of AR in the teaching and learning of more topics in secondary school mathematics, especially on the function and representation learning and the representational use in STEM subjects.

Additionally, the differences (3:7) among students in choosing between a simple game (AR Game 1) and a complex game (AR Game 2) justified our usage of scaffolding in the lesson design, especially since more low-achieving students preferred the easy game. By exploring the AR games in a simple-to-complex sequence, students could concentrate on the tasks within their range of competence first and then progress to the tasks beyond their ability, completing a "scaffolding" process [71]. Future designers should consider adding scaffolding elements in their designing process to support learners of varied abilities in accomplishing their learning goals.

Regarding recommendations for improving the AR app, students in our study suggested upgrading operation systems, diversifying game scenes, and extending playing time. Other studies, for example [72,73], also reported similar results with a few exceptions, especially regarding the collaborative learning approach. Students in our intervention explored the game independently as our AR app is not socially interactive. Researchers have recognized several benefits of collaborative learning for middle school mathematics students [74]. In the future, researchers can design a collaborated MRLE and see how social interactions influence students' learning sequence in MRLE. Additionally, in Pombo's study [73], many students cared about the size of the study group and the competition involved during the exploration. In addition, many empirical studies indicated that sufficient playing time facilitated conceptual change, for example [75] and multi-representational learning takes more time [27]. Therefore, we recommend that the initial attempt at creating such a collaborated MRLE could start with paired students, giving them a longer exploration time.

Second, the AR-based MRLE we designed for this study demonstrated its potential in promoting lower secondary school students' representational fluency in linear functions. This successful attempt to combine function and representation learning with AR encour-

ages future research on utilizing AR in the teaching and learning of more complicated and challenging mathematical topics.

Concerning students' learning preferences, AR outperformed the traditional learning environment because it was more interesting and helped to visualize and deepen students' understanding, affirming previous research findings, for example, ref. [49]. In addition, multi-RLE surpassed mono-RLE since it saved problem-solving time, provided a comprehensive knowledge structure, enhanced conceptual understanding, and clarified the confusion. Many students mentioned that they could compare different representations simultaneously in the MRLE, which was essential in strengthening their interpretations of linear functions. Researchers have shown that comparing and contrasting is an effective critical thinking strategy that helps to build students' memories, eliminate confusion, and improve student learning [76,77]. This research illustrated that an AR-based MRLE provided a satisfying platform for comparing and contrasting.

In this study, we exploited AR to add a real-life dimension to representational learning of linear functions, which often involved one or two representations, for example [19]. Our attempt to simultaneously teach students three representations (real-life, symbolic, and graphical) of linear functions was fairly successful. Since there are five common representations of functions, future research could (1) integrate more representations in an AR-based MRLE, for example, adding the tabular representation, and (2) investigate the maximum or appropriate number of representations that can be used in MRLE to support students representational learning effectively.

Third, students in this study exhibited different learning patterns during their interactions with representations in the AR-based MRLE. Specifically, high-achieving students (with stronger prior representational knowledge) tended to follow (1) concrete->semi-concrete->abstract and (2) proficient->non-proficient learning sequences, whereas low-achieving students demonstrated no explicit learning patterns. We can infer that high-achieving students could utilize AR well in facilitating their conceptual understanding, while low-achieving students were not capable of making the same progress.

Under such circumstances, teachers should take a more active role as more knowledgeable learners in guiding students to interpret the concepts within their zones of proximal development [71]. For instance, they could guide low-achieving students to start from concrete or their most proficient representations. Notably, generating a comprehensive and accurate representational learning profile would be a prerequisite for appropriate guidance.

Finally, regarding the theoretical contribution of our study, a conceptual framework to classify students' representational learning is proposed and used successfully. The Six Dimension Learning Profile score report (see Figure 4) could not only help students to enhance the connections that they are not quite proficient in but also support teachers to design and offer adaptive learning tasks targeting students' learning difficulties.

Furthermore, the conceptual framework we proposed is a further refinement of the Cartesian Connection (a point A is on the graph of Line L if and only if the coordinates of A satisfy the equation of line L) denoted by Moschkovich et al. [28]. Glen and Zazkis's [19] analysis elaborated a helpful additional facet of the Cartesian Connection, the slope facet, connecting visual and analytical representations of the slope. Our framework further extended the notion of the Cartesian Connection by (1) adding the *function as object* facet, that is, connecting the visual (graph of Line L) and analytical (algebraic expressions of Line L) representations, and (2) supplementing real-life representations, which could be an essential step in understanding linear functions.

Regarding limitations, given that this study was conducted in an urban area in Shandong with a convenient sample, the conclusions drawn from this work should be taken with prudence, which might not be generalizable to other parts of China or other countries. Future studies could consider a large randomly selected sample of students with different backgrounds and characteristics. In addition, some researchers pointed out that students in urban areas might employ AR in the classroom quickly even if it is their first touch with this technology compared to those in rural areas [58], as urban students have more

access to new technologies due to the digital divide [78]. Students in rural areas might encounter more cognitive loads in the AR-based MRLE. Future research could be conducted on comparing urban and rural students in their use of AR-based MRLE or exploring digital design features that support the rural students' learning with AR.

Additionally, as this study was conducted at the end of the semester, students had a busy schedule, and thus the intervention was shortened to two classes. We acknowledge it as a limitation of our intervention. Conceptual change takes time [75]. We hope that our research will inspire future long-term research in exploring (1) how students interact with different representations in the MRLE for a longer time, for example, one chapter or one semester, and (2) what kind of support is needed during a longer learning process. Such investigations may yield meaningful insights to cultivate students' representational fluency in daily mathematics teaching and learning.

## 5. Conclusions

The main objective of this research is to develop, implement, and evaluate an AR-based MRLE in helping students understand linear functions. In particular, we want to elucidate how students perceive and interact with an AR-based MRLE. The following are the primary findings of this research:

1. Overall, students are highly satisfied with the AR-based MRLE.
2. The AR-based MRLE can promote students' understanding of the real-life, symbolic, and graphical representations, individually, and connections among them of linear functions.
3. In the AR-based MRLE, high-achieving students demonstrate apparent learning sequences: from (1) concrete->semi-concrete->abstract and (2) proficient->non-proficient representations, whereas low-achieving students exhibit no explicit patterns in their interaction with representations.

Overall, students' learning in the multi-representational environment is a complex process influenced by the mathematics content, forms of representations, digital features, and students' representational learning profile, as we observed in this study. In sum, the contributions of this study can be seen in four aspects:

1. We extended the research on students' learning of linear functions and its three representations, using a well-designed AR-based MRLE.
2. We produced an initial attempt at examining how students interact with various representations in an MRLE via a qualitative approach.
3. We supported the idea that students' mathematical characteristics and abstract mathematics topics should be taken into consideration in future AR and mathematics education research with empirical evidence.
4. We refined the notion of the Cartesian Connection and proposed a conceptual framework to classify students' representational learning.

As this study aimed to encourage students to achieve a multi-representational learning goal, we intentionally excluded the option of hiding one or two representations when we designed the AR-based MRLE. Future research is recommended on how students learn in an MRLE with the flexibility to choose how many representations are visible to them via a qualitative design. Such a study could provide more information on students' thinking and how they choose or deal with each representation in the learning process, see also [27]. Moreover, the AR$^+$ Beijing travel plan game can be played by parents and children together at home, which could enhance family bonding and self-learning. Especially in the current COVID-19 pandemic, due to the paradigm shift, the issue of how to stimulate students' learning at home is of great importance and should be taken into consideration by future researchers.

**Author Contributions:** Conceptualization, S.L. and Y.S.; methodology, S.L., Y.S. and X.J.; software, X.J. and S.C.; validation, S.L., Y.S. and X.J.; formal analysis, X.J.; investigation, X.J.; data curation, S.L. and X.J.; writing—original draft preparation, S.L. and Y.S.; writing—review and editing, S.L., Y.S. and

S.C.; visualization, X.J.; supervision, S.C.; project administration, S.C.; funding acquisition, S.L. and S.C. All authors have read and agreed to the published version of the manuscript.

**Funding:** This research was in part supported by the National Natural Science Foundation of China (61977007).

**Institutional Review Board Statement:** Not applicable.

**Informed Consent Statement:** Not applicable.

**Data Availability Statement:** The data are not publicly available due to privacy restrictions.

**Acknowledgments:** The authors wish to thank Zifeng Liu, Yang Shen, Chen Kong, Li Li, and Ting Lu for their assistance at various stages of this study.

**Conflicts of Interest:** The authors declare no conflict of interest.

## Appendix A

Table A1 lists the codes we identified for answers to each interview questions.

**Table A1.** Final Codebook.

| # | Question | Code | Code Description |
|---|----------|------|------------------|
| Q1 | What do you think is the most impressive thing in an AR-based MRLE? | a<br>b | AR games<br>The motion problem |
| Q2 | Do you think AR helps you understand the real-life motion problem and its graph and algebraic expression better? | Y<br>N | Yes<br>No |
| Q3 | Which representation (the real-life motion problem, its graph and algebraic expression) that AR helps you understand the best? | RL<br>S<br>G | The real-life motion problem<br>The algebraic expression<br>The graph |
| Q4 | Do you pay attention to all three representations in the AR game interface simultaneously? | Y<br>N | Yes<br>No |
| Q5 | Will you pay attention to all three representations at the same time? Or do you focus on animation first, and then shift your attention to graphs or algebraic expressions? Or other paths? | L1<br>L2<br>L3<br>L4<br>L5<br>L6<br>L7<br>L8 | RL&G->S<br>RL->G->S<br>RL->G&S<br>RL&G&S<br>S->G->RL<br>G&S->RL<br>G->S->RL<br>G->RL&S |
| Q6 | What is the strength of the AR-based MRLE? | P1<br>P2<br>P3<br>P4<br>P5 | Demonstrate connections between different representations clearly<br>Promote a comprehensive structure of multiple representations<br>Easy to calculate/draw, and make the problem-solving process easier<br>Deepen the understanding of functions and function problems in real life<br>No strength |
| Q7 | What is the weakness of the AR-based MRLE? | Q1<br>Q2<br>Q3 | Feel it hard to concentrate on three representations simultaneously<br>Pay too much attention to the RL animation<br>No weakness |

**Table A1.** *Cont.*

| # | Question | Code | Code Description |
|---|----------|------|-----------------|
| Q8 | Do you prefer multi- or mono-representational learning environment? Why? (Text analysis for Why?) | H | Multi-representational |
| | | I | Mono-representational |
| Q9 | Do you prefer AR-based or traditional non-AR learning environment? Why? (Text analysis for Why?) | AR | AR-based |
| | | Non-AR | Traditional non-AR |
| Q10 | Which AR games (AR Game 1 or Game 2) do you like better? Why? (Text analysis for Why?) | AR1 | AR Game 1: Let's go hiking |
| | | AR2 | AR Game 2: AR$^+$ Beijing travel plan |
| Q11 | Overall, are you satisfied with the whole intervention class? | SS | Extremely satisfied |
| | | S | Satisfied |
| Q12 | Any suggestions? (Text analysis) | / | / |

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
