# Peer review of "Using Augmented Reality to Enhance Students’ Representational Fluency: The Case of Linear Functions"

_mathematics, doi:10.3390/math10101718_

Round 1

Reviewer 1 Report

This study attempts to determine whether the medium of Augmented Reality (AR) enhances the understanding that students can develop in algebra, particularly for linear functions. Through this study the authors do find that multiple representations of linear functions were better understood via AR

The paper is well-written. The literature review is sufficient and the current work is also justified quite well. Good analysis has been done along with an in-depth discussion about the observations and the reasons behind them.

I would recommend acceptance of the paper in its current form.

Author Response

Response to reviewer 1

(Manuscript ID: mathematics-1697891)

Dear reviewer,

We really appreciate your feedback. We are very grateful for your comments on the manuscript. All of your comments were answered one by one.

Here are responses to the comments from reviewer 1:

  1. This study attempts to determine whether the medium of Augmented Reality (AR) enhances the understanding that students can develop in algebra, particularly for linear functions. Through this study the authors do find that multiple representations of linear functions were better understood via AR.

Reply: Thanks for your comments. We hope that our research will attract more attention to AR-based multi-representational learning environments.

  1. The paper is well-written. The literature review is sufficient and the current work is also justified quite well. Good analysis has been done along with an in-depth discussion about the observations and the reasons behind them.

Reply: Thank you very much for your appreciation.

  1. I would recommend acceptance of the paper in its current form.

Reply: Thank you very much for your appreciation.

Reviewer 2 Report

This is an interesting study and carefully performed with useful results. I have provided a small set of adjustments to the use of English that the authors should consider seriously. I have marked the attached file in yellow and made comments with notes on the pdf. 

Author Response

Response to reviewer 2

(Manuscript ID: mathematics-1697891)

Dear reviewer,

We really appreciate the detailed feedback from you. The manuscript has been revised according to your insightful comments and valuable suggestions. We hope this revised version has addressed all your concerns.

Here are responses to the comments from reviewer 2:

  1. This is an interesting study and carefully performed with useful results. I have provided a small set of adjustments to the use of English that the authors should consider seriously. I have marked the attached file in yellow and made comments with notes on the pdf.

Reply: Thank you very much for your detailed 32 comments regarding the use of English. According to your advice, we amended the relevant part (marked in yellow) in the manuscript, and explained the adjustments one by one as follows.

  • On page 1, line 10, we changed to “multiple representations of linear functions”.
  • On page 1, line 11, we changed to “these teaching” as you suggested.
  • On page 1, line 12, we changed to “our study” as you suggested.
  • On page 1, line 33, we changed to “between various representations” as you suggested.
  • On page 1, line 42, we adjusted to “convenient for graphing and calculating” as you suggested.
  • On page 2, line 74, we amended the original one to “the effect of using Diagram” as you suggested.
  • On page 3, line 112, we added the full name of ITPC: “based on the Integrated Model of Text and Picture Comprehension (ITPC)”.
  • On page 3, line 114, we changed to “representations occur” as you suggested.
  • On page 3, line 122, we adjusted to “a student’s representational fluency” as you suggested.
  • On page 3, line 123, we amended it to “a student’s prior knowledge” as you suggested.
  • On page 4, line145, we added “packages” as you suggested.
  • On page 4, line 189, we added “study” as you suggested.
  • On page 5, line 213, we changed to “students’ learning processes” as you suggested.
  • On page 5, line 225, we adjusted to “a typesetting issue” as you suggested.
  • On page 7, line 251, we accepted your insightful advice and changed the original one to “make the mathematic easier.”
  • On page 8, line 293, we adjusted to “assess reliability” as you suggested.
  • On page 9, line 321, we changed to “questionnaires” as you suggested.
  • On page 9, line 332, we adjusted to “detailed learning profile” as you suggested.
  • On page 10, line 360, we deleted the sentence as you suggested.
  • On page 11, line 384, we added “a” as you suggested.
  • On page 12, line 439, we changed to “students’” as you suggested.
  • On page 13, line 452, we accepted your advice and changed it to “a linear functions’ graph with a direct variations’ equation”.
  • On page 14, line 507, we deleted “on” as you suggested.
  • On page 14, line 544, we added “RLE” as you suggested.
  • On page 15, line 552, we changed to “add a real-life dimension to representational learning” as you suggested.
  • On page 15, line 571, we deleted “the” as you suggested.
  • On page 15, line 585, we deleted “the” as you suggested.
  • On page 15, line 539, we changed to “compared to” as you suggested.
  • On page 16, line 627, we added “the idea” as you suggested.
  • On page 17, line 634, we changed to “Future research is recommended on …” as you suggested.
  • On page 17, line 639, we adjusted it to “parents and children” as you suggested.
  • On page 17, line 641, we added “and” as you suggested.

Again, many thanks for your helpful advice.

Reviewer 3 Report

This work aims at investigating an augmented reality based multi-representational learning environment to enhance student representational fluency on the subject of linear functions. The authors focus on specific applications in the uniform linear motion problem in secondary school physics and mathematics and analyse data collected from 82 students from two high performing classes in an urban area in China.
Although, the results are not fully conclusive and more research is needed to provide additional  information on these issues, the subject is interesting, the article is clear and well written, and the analysis is detailed and well organized. In view of the above, I recommend this article for publication to Universe.

Author Response

Response to reviewer 3

(Manuscript ID: mathematics-1697891)

Dear reviewer,

We really appreciate the feedback from you. We are very grateful for your appreciation of the manuscript.

Here are responses to the comments from reviewer 3:

  1. This work aims at investigating an augmented reality based multi-representational learning environment to enhance student representational fluency on the subject of linear functions. The authors focus on specific applications in the uniform linear motion problem in secondary school physics and mathematics and analyse data collected from 82 students from two high performing classes in an urban area in China. Although, the results are not fully conclusive and more research is needed to provide additional information on these issues, the subject is interesting, the article is clear and well written, and the analysis is detailed and well organized. In view of the above, I recommend this article for publication to Universe.

Reply: Thank you so much for your appreciation.